# Artificial Intelligence and Corneal Confocal Microscopy: The Start of a Beautiful Relationship

**DOI:** 10.3390/jcm11206199

**Published:** 2022-10-20

**Authors:** Uazman Alam, Matthew Anson, Yanda Meng, Frank Preston, Varo Kirthi, Timothy L. Jackson, Paul Nderitu, Daniel J. Cuthbertson, Rayaz A. Malik, Yalin Zheng, Ioannis N. Petropoulos

**Affiliations:** 1Department of Cardiovascular & Metabolic Medicine, Institute of Life Course and Medical Sciences, University of Liverpool, Liverpool L69 3BX, UK; 2Division of Diabetes, Endocrinology and Gastroenterology, Institute of Human Development, University of Manchester, Manchester M13 9PL, UK; 3King’s Ophthalmology Research Unit, Faculty of Life Sciences and Medicine, King’s College London, London SE5 8AB, UK; 4Weill Cornell Medicine-Qatar, Doha P.O. Box 24144, Qatar

**Keywords:** artificial intelligence (AI), deep learning algorithm (DLA), corneal confocal microscopy (CCM), corneal nerve fractal dimension (CNFrD)

## Abstract

Corneal confocal microscopy (CCM) is a rapid non-invasive in vivo ophthalmic imaging technique that images the cornea. Historically, it was utilised in the diagnosis and clinical management of corneal epithelial and stromal disorders. However, over the past 20 years, CCM has been increasingly used to image sub-basal small nerve fibres in a variety of peripheral neuropathies and central neurodegenerative diseases. CCM has been used to identify subclinical nerve damage and to predict the development of diabetic peripheral neuropathy (DPN). The complex structure of the corneal sub-basal nerve plexus can be readily analysed through nerve segmentation with manual or automated quantification of parameters such as corneal nerve fibre length (CNFL), nerve fibre density (CNFD), and nerve branch density (CNBD). Large quantities of 2D corneal nerve images lend themselves to the application of artificial intelligence (AI)-based deep learning algorithms (DLA). Indeed, DLA have demonstrated performance comparable to manual but superior to automated quantification of corneal nerve morphology. Recently, our end-to-end classification with a 3 class AI model demonstrated high sensitivity and specificity in differentiating healthy volunteers from people with and without peripheral neuropathy. We believe there is significant scope and need to apply AI to help differentiate between peripheral neuropathies and also central neurodegenerative disorders. AI has significant potential to enhance the diagnostic and prognostic utility of CCM in the management of both peripheral and central neurodegenerative diseases.

## 1. Introduction

Peripheral neuropathies are highly prevalent neurological disorders which result in motor, sensory and autonomic disturbances, but can pose significant diagnostic challenges [1]. Their epidemiology is dependent on patient population and diagnostic definitions; however, they affect ~3–8% of the general population, with an increasing prevalence with age [2]. Globally, the most common causes are diabetes mellitus, chemotherapy-induced peripheral neuropathy (CIPN) and human immunodeficiency virus infection (HIV), which result in a length-dependent neuropathy [3]. Diabetic peripheral neuropathy (DPN) is present at diagnosis in ~8% of people with diabetes [4], with an increasing prevalence with longer diabetes duration, reaching 30–66% in unselected populations [5,6]. Sensory neuropathy associated with HIV affects up to 35% as a direct consequence of the viral infection, or as a consequence of toxicity from antiretroviral therapy [7]. CIPN can lead to the cessation or discontinuation of chemotherapy [8] and can affect ~70% of patients receiving oxaliplatin [9].

The diagnosis of peripheral neuropathies has relied heavily on identifying neuropathic symptoms and neurological deficits on neurological examination [10]. Questionnaires for symptoms and composite scores derived from symptom and signs allow a relatively quick and cost-effective approach in the clinic with minimal training, but they are imprecise [11]. Quantitative sensory testing quantifies dysfunction of both small and large nerve fibres but is limited by poor reproducibility and inability to identify the site of the lesion and underlying pathophysiology [12,13]. Evaluation of intra-epidermal nerve fibre density (IENFD) is a reliable means to assess small fibre neuropathy [14] and IENFD loss occurs in pre-diabetes [15] and predicts the onset of clinical neuropathy [16]. Nerve regeneration has also been reported after successful lifestyle modification [17,18]. However, the requirement for an invasive skin biopsy, risk of infection at the biopsy site and lack of laboratories with expertise to quantify IENFD limits its use. Nerve conduction studies (NCS) are used extensively and are considered the reference standard method for the diagnosis of peripheral neuropathies, especially in differentiating between axonal and demyelinating disorders [19]. However, they cannot identify small fibre involvement in mixed and pure small fibre neuropathies as they primarily assess large, myelinated (β) nerve fibres. Artificial intelligence (AI) is already established in the screening of diabetic retinopathy and AI-based deep learning algorithms (DLA) have consistently demonstrated accurate corneal nerve segmentation of common peripheral neuropathies.

## 2. Corneal Confocal Microscopy

### 2.1. Corneal Anatomy

The human cornea is the most densely innervated tissue in the human body, receiving sensory innervation from the ophthalmic branch (V1) of the trigeminal ganglion. The trigeminal ganglion contains approximately 27,400 pseudo-unipolar neurons of which ~1.3% innervate the cornea [20]. Nerve bundles from V1 penetrate the cornea at the level of mid-stroma where their myelin sheath is lost. Following a complex route branching into several smaller-diameter nerves, they reach the corneal surface to form the sub-basal nerve plexus, which is located between the Bowman’s layer and the basal epithelium and consists entirely of unmyelinated C-fibres [21]. A single neuron supports 200–3000 corneal nerve endings, corresponding approximately to 11,000 fibres/mm^2^ of the corneal surface [15]. These C-fibres are powerful co-regulators of the corneal homeostasis and are key to providing protection from external stimuli through polymodal nociceptors (70%), mechano-receptors (20%) and thermal receptors (10%) [22]. Early reports from ~40 years ago suggested a loss of corneal sensation in type 1 diabetes [23].

### 2.2. Corneal Confocal Microscopy

CCM is a rapid, non-invasive, and reiterative technique to image the cornea at 600× magnification in the clinic, without the need for expensive or time-consuming laboratory analysis. CCM has been extensively used in ophthalmology practice to diagnose and manage corneal disease and to monitor wound healing following surgery. Over the past 20 years, our group has pioneered the use of CCM as a biomarker of peripheral neuropathy. In 2003, we demonstrated that CCM identifies subclinical and progressive loss of corneal sub-basal nerves with increasing severity of diabetic neuropathy [24]. Subsequently, Quattrini et al. [25] demonstrated comparable reductions in IENFD and corneal nerve measures in DPN. The subsequent transition from a white light to a laser-based CCM enhanced the scanning resolution and improved our ability to detect nerve abnormalities and patterns associated with the underlying pathological process [26,27,28].

## 3. The Diagnostic Efficacy of Corneal Confocal Microscopy in Peripheral Neuropathies

CCM has a growing body of evidence demonstrating that corneal nerve fibre loss is a validated surrogate biomarker of peripheral neuropathies and central neurodegenerative diseases [29,30]. CCM has age-adjusted normative values in populations from the US, Europe, and Australia [31]. Dehgani et al. [32] demonstrated the stability of the sub-basal nerve plexus in healthy volunteers over 36 months. The overall accuracy and diagnostic efficacy of CCM has been extensively studied in DPN (Table 1) [33]. CCM has consistently shown greater corneal nerve fibre loss in patients with DPN compared to without DPN and healthy controls, demonstrating good-to-excellent sensitivity and specificity [24,25,34,35,36,37,38,39]. Both Alam [40] and Chen et al. [41] demonstrated that CNFD had superior performance compared to IENFD for the diagnosis of DPN. A multicentre, multinational NIH consortium study of 998 participants with type 1 and type 2 diabetes reported that CNFL had a 0.88 sensitivity and 0.88 specificity for the diagnosis of DPN [42]. A recent study (*n* = 220) using a higher optimal threshold than the consortia paper (CNFL < 15.3 mm/mm^2^) reported 0.8/0.59 sensitivity/specificity for diagnosing DPN [43]. CNFL predicts incident neuropathy over 4 years [25,26], thus providing a putative ‘at-risk’ threshold. Indeed, rapid decline of corneal nerves (defined as >−6% annual CNFL loss) can stratify patients at greatest risk for DPN onset and progression [44] recently been confirmed by Alam et al. [45]. Our data have also shown that the risk factors for corneal nerve loss differ between type 1 and type 2 diabetes, with an association with LDL cholesterol and triglycerides in type 1 and age, glycemia and weight in type 2 diabetes [46], especially in patients with a more rapid nerve fibre decline [44,45].

## 4. Beyond Diabetic Peripheral Neuropathy

### 4.1. CCM in Other Peripheral Neuropathies

CCM has been used to identify corneal nerve loss in a range of peripheral neuropathies including idiopathic small fibre neuropathy (ISFN), CIPN, chronic inflammatory demyelinating polyneuropathy (CIDP), HIV-associated neuropathy, hereditary sensory-motor neuropathies [26,51,52,53], Fabry’s disease [54,55], hypothyroid neuropathy [56], amyloid neuropathy [48,57], inflammatory neuropathies [58,59,60], and fibromyalgia [61,62]. The extent of corneal nerve loss has been associated with underlying pathophysiological defects and the severity of symptoms and deficits. In HIV-associated neuropathy, CNFD correlated with neuropathic symptom burden [53], whilst CNFD and CNFL correlated with the neurological component of the Mainz Severity Score Index (MSSI) and α-galactosidase A enzyme activity in patients with Fabry’s disease [54]. CNFD, CNFL and CNBD have also been shown to correlate with neurological deficits assessed using the Scale for the Assessment and Rating of Ataxia (SARA) and Friedreich’s Ataxia Rating Scale (FARS) as well as the genotype (GAA repeats) [63]. Recently, CNFL and inferior whorl length have demonstrated superior sensitivity and specificity in identifying participants with transthyretin familial amyloid polyneuropathy [48].

### 4.2. CCM in Central Neurodegenerative Disease

Corneal nerve pathology has been demonstrated in patients with Parkinson’s disease, multiple sclerosis (MS), dementia [64,65,66] and amyotrophic lateral sclerosis (ALS) [67]. Corneal nerve fibre loss is associated with cognitive decline and correlates with functional independence in mild cognitive impairment and dementia [68]. In MS, CNFD is consistently reduced [65,69,70]. Petropoulos et al. [70] and Bitirgen et al. [65] have additionally reported a reduction in CNFL and CNBD. In one study, dendritic cell density increased [65], whilst in another study there was no difference in MS [69]. The heterogeneity of findings in patients with MS makes it challenging to arrive at a diagnosis based solely on CCM. However, AI-based analysis of CCM images has shown good diagnostic utility in different subtypes of MS [71]. CNFL and CNFD are both reduced in ALS when compared to healthy controls and correlate with a worsening ALS-functional rating scale [67]. In Parkinson’s disease, motor abnormalities and cognitive dysfunction correlated with corneal nerve fibre loss [64,72]. A recent study by Che et al. [49] has demonstrated that CNFD has an area under the curve (AUC) of 0.96 with sensitivity/specificity of 0.95/0.88 in identifying patients with Parkinson’s disease. Alterations in corneal innervation precede loss of IENFD in Parkinson’s disease [73] Changes in the number of corneal nerve bifurcations and beading [66] may be a possible biomarker of early disease. The presence of corneal nerve abnormalities in multiple peripheral neuropathies and central neurodegenerative diseases, especially in elderly populations with multiple comorbidities limits specificity for diagnosis, necessitating perhaps alternative metrics [74]. As such, there is increasing interest in quantification of the complexity of corneal nerve topography and end-to-end classification with DLA to help differentiate different peripheral neuropathies and central neurodegenerative diseases.

## 5. CCM Image Acquisition and Analysis

Non-overlapping (<20%) images are acquired from the corneal apex (central cornea) and 5–8 images are selected based on the depth and quality to ensure accurate and representative analysis [75]. Quantitative analysis of corneal nerves can be undertaken using manual (CCMetrics, The University of Manchester, Manchester, UK) and automated (ACCMetrics, The University of Manchester, Manchester, UK) software [76]. Both approaches rely on nerve segmentation and quantification of different sub-basal nerve plexus metrics [39]. These metrics include corneal nerve fibre length (CNFL), corneal nerve fibre density (CNFD), and corneal nerve branch density (CNBD). CNFL is regarded as the primary biomarker showing an early reduction in diabetes [77], and it is also the most reproducible corneal nerve metric [78]. Together with CNFD and CNBD, they make up three key CCM parameters, that enable an assessment of nerve degeneration and regeneration (Table 2) [79]. Other automated measures which are less well validated include corneal nerve fibre tortuosity (CNFT) [80], corneal nerve fibre total branch density (CTBD), corneal nerve fibre area (CNFA), corneal nerve fibre width (CNFW) [81], nerve fibre beading [82], fractals [74] inferior whorl length [79] corneal nerve connection points, and average weighted corneal nerve fibre thickness [83]. Figure 1 illustrates the difference between primary corneal nerve fibres and their branches.

### 5.1. Manual Analysis

Manual analysis relies on expert manual annotation and tracing of corneal nerves to delineate main nerve fibres, branch points and fibres. Dabbah et al. [84] developed CCMetrics (The University of Manchester, Manchester, UK), a purpose-built interactive image analysis software to facilitate manual quantification of CNFD, CNBD, CNFL and CNFT in each CCM image. The narrow field of view of individual images has been perceived as a limitation, as such, some centres have used wide field imaging to create sub-basal nerve plexus maps [85]. Participants are asked to focus on a number of dots across a square grid in a systematic process, with images acquired mapping out the corneal sub-basal nerve plexus and inferior whorl [78]. Whilst manual annotation has demonstrated reliability and reproducibility [30,86], it is labour intensive and requires considerable expertise [81,84]. For CCM to be utilised as a clinically useful diagnostic tool, reliable automated analysis is required [84], and this has led to the development of automated CCM image analysis software [81,84].

### 5.2. Automated Analysis

Automated CCM image analysis software evaluates nerve fibres through segmentation and quantification of corneal nerve morphological features. Dabbah et al. [87] developed a multi-scale, adaptive, dual-model detection algorithm, combining a background (noise and underlying connective tissue) Gaussian model and a foreground (corneal nerve fibres) Gabor wavelet-based model, which achieved a strong correlation with manual-ground truth annotations. Dabbah et al. [84] published the deterministic software, ACCMetrics, a multi-scale dual model with a neural network pixel classification-based algorithm that automatically quantifies CNFD, CNBD, CNFL, CNFA and CNFW and shows an excellent correlation with manual ground truth. Petropoulos et al. validated ACCMetrics by demonstrating significant reductions in CNBD, CNFD and CNFL which correlated with increasing neuropathic severity in DPN [39]. Subsequently, Chen et al. [81] demonstrated comparable efficacy between ACCMetrics and IENFD in diagnosing DPN. Recent advances in the automated analysis of CCM images have utilised DLA [88,89,90] and abandoned nerve segmentation in favour of end-to-end classification to allow the DLA to determine features of importance for image classification [91,92].

## 6. Fractal Dimension

Complex geometric structural properties can be characterised by different scale values, namely fractal geometry [93]. A fractal dimension is a mathematical parameter that describes the complexity of a biological structure within a two- or three-dimensional space [94]. Fractal dimension have been used in medical imaging to investigate retinal fundus images in a variety of ocular diseases.

### 6.1. Fractal Dimension in Diabetic Retinopathy

The architecture of the retinal microvasculature has been quantified using fractal analysis in patients with diabetes [95,96,97]. Cheung et al. [95] reported that increased fractal dimension, representing increased complexity of the retinal microvasculature, was associated with higher odds of developing diabetic retinopathy (DR), independent of HbA1c. However, Fan et al. [97] reported no difference in fractal dimension in people with DR compared to controls. Talu et al. [96] demonstrated that fractal dimension were higher in mild non-proliferative DR (NPDR), but lower in moderate and especially severe NPDR when compared to controls. Torp et al. [98] demonstrated that retinal venular fractal dimension predicted disease activity 6 months after panretinal photocoagulation, suggesting it may serve as a biomarker of treatment efficacy. The challenges of utilising the fractal dimension in DR have been described by Forster et al. [99]. Thus, with the progression of DR, different vascular changes can result in opposing fractal indices, with an initial decrease followed by increased fractal dimensions in proliferative DR [100]. Given the variability in fractal dimension in different stages of DR, they have not been incorporated in diagnostic tools and protocols. More research is required on the evolution of fractal dimension in DR in relation to disease severity and treatment in well-designed natural history studies. Spectral-domain optical coherence tomography angiography (OCTA) generates detailed anatomical images of the superficial and deep capillary plexus. Zahid et al. [94] showed that OCTA derived fractal dimensions were reduced in DR when compared to controls, but there was no further analysis in relation to the grade of retinopathy. Current DR detection DLA do not use explicitly extracted features, such as fractal dimensions, but instead learn a unique set of features directly from fundal images [101].

### 6.2. Fractal Deimention in Corneal Confocal Microscopy Images

Corneal nerve geometry can be quantified using corneal nerve fractal dimension (CNFrD) [74]. CCM images are divided by two-dimensional square grids to generate boxes subtended by a corneal nerve segment. When a nerve fibre is detected within a box, this box is further subdivided into equally sized, smaller boxes and the process is repeated until no nerves are detectable. Thus, a higher number of boxes signifies greater pattern complexity, generating a higher fractal dimension index [26,74].

The complexity of corneal nerve architecture is altered in DPN [102]. The diagnostic utility of CNFrD was first assessed by Chen et al. [74], comparing healthy controls to participants with type 1 diabetes, with or without DPN. Automated CNFrD (AUC 0.74) had a comparable diagnostic efficacy to automated CNFL (AUC 0.74) and CNFD (AUC 0.77), suggesting that CNFrD may serve as an additional metric for identifying DPN. Petropoulos et al. [26] also demonstrated that CNFrD detected differences in corneal nerve topography by aetiology of underlying neuropathy, even when adjusting for CNFL. Patients with CIDP, HIV-SN, CIPN and DPN had a lower CNFrD when compared to healthy controls, but CNFrD was significantly lower in DPN compared to CIDP, HIV-SN and CIPN. To date no studies have sought to assess the utility of CNFrD in discriminating between peripheral and central neurodegenerative diseases, perhaps by combining CNFL, CNFD, CNBD with CNFrD utilising artificial intelligence (AI).

## 7. Artificial Intelligence (AI)

Artificial intelligence (AI) is a major field of study that aims to develop computers and machines that can emulate aspects of human intelligence. Machine learning, a sub-field of AI, describes a large and diverse set of algorithms that learn through examples and data. Artificial neural networks (ANN) are machine learning algorithms structured much like biological neurons. ANNs can be multi-layered to improve their performance on complex tasks which is termed deep learning. The non-invasive nature of CCM, and the ability to capture a large quantity of images leaves it well positioned for use in the AI revolution for data-driven disease modelling.

### 7.1. Artificial Intelligence and Deep Learning

AI aims to automate complex tasks normally performed by humans and is best understood as an umbrella term, with DLA being a specific type of AI. DLA are multi-layered artificial neural networks (ANN) whose architecture is inspired by biological neurons [103]. Each ANN contains nodes (akin to cell bodies) that communicate with other nodes via connections (akin to neural axons). An increase in the number of nodes between the input layer (akin to receptors) and output layer increases the complexity of the network, and its capacity to learn and perform complex tasks. A common way to train ANNs is through a process called supervised learning, whereby the network is given the correct answer (ground truth) for each case, e.g., the correct neuropathy category for a given CCM image. Utilising multiple images, the ANN iteratively learns which connections between nodes to strengthen or weaken so that, for a given input, it generates a prediction which is as close as possible to the ground truth, and which best minimises the prediction error [104]. Specialised ANN adept at analysing images are convolutional neural networks (CNN) which have filter-like components which can localise and extract image features [105].

### 7.2. Artificial Intelligence and Opthalmology

Ophthalmology lends itself well to AI due to the diverse range of digital imaging modalities and large datasets produced during standard clinical care. DR, age related macular degeneration (ARMD), retinopathy of prematurity (ROP), glaucoma and cataracts have all been identified as promising disease areas for AI [106]. A systematic review of AI in the management of people with cataracts found that AI-driven diagnosis was at least comparable, and at times superior to expert clinical diagnosis [107]. AI has demonstrated utility beyond diagnostic purposes with improved intraoperative lens selection and a subsequently reduced refractive error [108]. AI algorithms have additionally demonstrated equivocal detection of ARMD when compared to ophthalmologists [109], and have outperformed optical coherence tomography metrics in differentiating between glaucomatous and healthy eyes [110]. AI has also been applied to detect DR and a DLA has gained formal approval by the US FDA for the detection of more-than-mild DR (IDx-DR) (sensitivity 87.2%; specificity 90.7%) [111,112].

### 7.3. Artificial Intelligence and Diabetic Retinopathy Screening

With the increasing worldwide prevalence of diabetes and DR [113], solutions to mitigate the human capital and associated health economic costs of expert graders to identify early DR in retinal fundus, enabling prompt diagnosis and treatment to prevent visual loss from DR [114] However, if the UK DR screening model was applied worldwide, by 2045, ~2 to 3 billion retinal images per year would require human grading [115]. AI has the potential to augment clinical decision making and reduce physician/assessor burden [116]. In non-AI-utilising centres/screening programmes, primary assessors review and grade retinal images and refer cases of suspected DR to secondary assessors for further analysis [117]. Currently in the UK, only Scotland incorporates automated AI detection of DR in their eye screening service [112]. iGradingM (version 1.1, Medalytix/EMIS health, Leeds, UK) is an automated retinal image analysis system used by the Scottish screening programme which replaces the primary human assessor, reducing the grading workload because only images with suspected DR are further graded by the secondary human assessor who decides whether patients are to be invited back for screening or referred to the hospital eye service for vision-threatening DR [118]. The use of AI as the primary assessor is cost-effective [119] and reduced manual grading workload by >35% [120], whilst achieving a sensitivity of 97.8% for referable DR [120]. EyeArt, another commercially available software, was recently evaluated in the English DR screening programme, and demonstrated a sensitivity of 96% for referrable DR with a specificity of 68% for no DR, but has yet to be implemented in England [115]. Takahashi et al. [121] demonstrated a DLA (modified GoogLeNet CNN) which can provide prognostic information and determine future treatment for DR. Indeed, DLA have demonstrated accurate prediction for the progression of DR from no to mild or worse disease over a 2-year time interval using fundus images (AUC: 0.79) [122]. Therefore, AI has the potential to be used within a personalised hybrid model for more individualised screening and treatment of DR.

## 8. AI in CCM

### 8.1. Technical Aspects of AI in CCM

The success of AI is underpinned by the availability of big data, high-performance computing resources such as graphics processing units (GPU), and open-source development libraries (e.g., PyTorch™ and Tensorflow™). DLA, such as CNNs, have demonstrated comparable, and at times superior performance compared to human experts in a plethora of applications. Hence, DLA have been developed and validated to analyse CCM images.

The non-uniform illumination or imbalanced intensity in CCM images poses challenges in evaluation for the diagnosis of diseases. Ma et al. [123] investigated the use of a generative adversarial network (GAN) to enhance CCM images, which utilise a cycle structure and illumination constrained GAN combining the benefits of global adversarial loss and cycle consistency constraints. The proposed generator and discriminator pairs consistently improve the image quality and were trained together, whereby the ‘generator’ translates images from low to high quality until the discriminator model cannot distinguish between the original and enhanced images by more than chance (equal to 50% accuracy). Williams et al. [90] proposed to segment corneal nerves from CCM images using a U-Net model [124] and used an ensemble of multiple U-Net models to demonstrate the model’s superior segmentation and classification performance compared to the widely available ACCMetrics [81]. Mou et al. [125] developed a corneal nerve fibre segmentation model based on a dual attention (e.g., spatial and channel attentions) mechanism for the prediction of regions of interest and demonstrated the model’s effectiveness through their automated DLA that demonstrated significant differences in the tortuosity of nerves between patients with diabetes and healthy controls [124]. Preston et al. [92] developed a DLA using ResNet [126] as the backbone network to diagnose peripheral neuropathy (in diabetes and prediabetes), achieving a high level of classification accuracy using end-to-end classification (Figure 2). They also demonstrated image attribution-based explainability methods to produce ‘heatmaps’ showing areas in the image which were important to the classification prediction (Figure 3).

### 8.2. AI Models in CCM

Standard CNN models [127] will typically take one CCM image per patient as the input, and extract the semantic features (features that help the model distinguish between different pathologies), via passing them through multiple convolution operations of different CNN layers. The semantic features are key to the DLA decision-making process. The convolution operation is regarded as a sliding window that filters along the whole image’s intensity values, resulting in a distilled feature map. After iterative convolution operations, the semantic features are summarised at a higher-level through pooling operations, which result in the final output (presence or absence of disease).

For CCM image acquisition, each patient has multiple images taken, and depending on patient factors, the number of images per patient may vary. Some images contain discriminative and trustworthy information that can be used for classification of disease, while others do not. Therefore, it is critical to automatically select the trustworthy and discriminative images which will contribute to the model’s final prediction.

CA-MIL (Consensus-Assistant Multiple Instance Learning) can take an arbitrary number of CCM images per patient as it’s input. This can strengthen the upper-bound ability of the DLA model, as more input information is assessed during the feature learning and extraction process.

There is a consensus-assistant module that verifies the reliability of selected CCM images. Random noise (such as Gaussian noise) is inserted into the model along with the input CCM images to adversely perturb the model’s feature learning and prediction process. In other words, the predictions of a generalisable MIL classifier should be robust enough not to be influenced by input perturbations, and if the predicted class score changes significantly under a certain perturbation, this suggests an unreliable CCM image [128].

This differs from classic MIL-based methods [129] that treat every CCM image per patient equally. These models are more liable to be biased by images without discriminative and trustworthy features, leading to a less reliable feature learning process.

### 8.3. AI in Diabetic Neuropathy

Table 3 summarises studies which have utilised AI in the analysis of CCM images. Williams et al. [90] demonstrated their DLA achieved an AUC of 0.83, and specificity/sensitivity of 0.87/0.68, respectively, for the detection of DPN. Salahouddin et al. [130] demonstrated that their image segmentation DLA had excellent correlation with manually quantified CNFL, and outperformed ACCMetrics with an excellent AUC (1.0), sensitivity (1.0) and specificity (0.95) for differentiating between healthy controls and patients with DPN. The model also achieved an excellent AUC (0.95), sensitivity (0.92) and specificity (0.8) for discriminating between patients with and without DPN. Scarpa et al. [89] used a CNN to simultaneously analyse multiple CCM images and achieved a 96% classification accuracy to differentiate people with DPN from controls, but there was no comparison between people without DPN. The DLA developed by Preston et al. [92] reported a sensitivity of 1.0, 0.85 and 0.83 for correctly identifying healthy controls, DPN− and DPN+, respectively, by utilising only one image per participant, and without segmentation prior to classification (end-to-end classification) [131]. To date, AI has been trained and validated on relatively small datasets [89,90,92,130,131]. Current AI models for the analysis of CCM images and diagnosis of DPN have achieved promising results, but large scale, external and prospective clinical validation is required [92]. Studies on large representative ‘real world’ datasets are critical for developing and evaluating AI algorithms that gain FDA approval for clinical application. In addition, CCM will require regulatory approval of its use as a medical device for screening and diagnosis of diabetic and other peripheral neuropathies as well as central neurodegenerative diseases [132].

Considering the mass adoption and regulatory perspective, the ‘black-box’ nature of AI has been subject to resistance by some in the medical community [133]. By the nature of analysing whole images and discriminating patterns, and not classifying based on specific abnormalities, AI’s lack of ‘explainability’ has raised perhaps legitimate concerns in the medical community [134]. Only a limited number of studies utilising AI and CCM have sought to provide class activation maps (or heat maps). Such attribution maps attempt to highlight the influence of different regions of the image on the final diagnostic output of the AI-based DLA [92]. The maturation of such tools is required to overcome some of the current barriers to clinical use of AI. The current literature undoubtedly supports the potential of AI in automating the analysis of CCM images to detect DPN. The ability to consistently distinguish patients with and without DPN, and subclinical DPN, underpins the power of DLA [92]. The accurate and precise interpretation of CCM images by AI has the potential to lead to a paradigm shift in the diagnosis of various neurodegenerative diseases.

## 9. Future Clinical Applications

CCM identifies small nerve fibre damage in early DPN [135,136] before an abnormality in currently accepted endpoints, such as symptoms and deficits and nerve conduction studies. Future studies should utilise AI in CCM and AI to differentiate different peripheral and central neurodegenerative diseases using end-to-end classification and in-depth analysis of corneal topography/geometry. This may be especially useful at first presentation of neurological disease; often where diagnostic doubt exists. We have previously shown that simultaneous pancreas and kidney (SPK) transplantation in patients with type 1 diabetes is associated with corneal nerve regeneration after 6 months followed by an improvement in neuropathic symptoms after 24 months and nerve conduction after 36 months [137] within 28 days of blocking inflammation with ARA-290 in sarcoidosis [138,139]. We have also shown evidence of corneal nerve regeneration after bariatric surgery in obese subjects with [140] and without diabetes [141]. In a randomised clinical trial, weekly GLP-1 with pioglitazone or basal bolus insulin led to a~3% improvement in HbA1c, which was associated with corneal nerve regeneration over 12 months, but with no change in vibration perception or sudomotor function [142]. Two recent trials with omega-3 fatty acid in patients with type 1 diabetes have demonstrated corneal nerve regeneration with no change in nerve conduction velocity, thermal thresholds, or autonomic nerve function [143,144]. Thus, CCM truly satisfies the FDA criteria for a biomarker and could play an important role as an endpoint in clinical trials of therapies for diabetic and other peripheral neuropathies, as well as central neurodegenerative diseases [145]. CCM has the potential to adopt the hub and spoke model currently employed the UK’s national diabetic retinopathy screening programme, whereby images are collected locally and interpreted at a central hub by AI. This method reduces intra-regional variation and standardises the analytical process. Such a hub and spoke model allows large volumes of data to be collated, improving the output and accuracy of future models.

## 10. Conclusions

AI has the potential to fully unleash the full ability of CCM for the diagnosis of DPN, peripheral neuropathies and other neurodegenerative diseases. The incorporation of novel corneal nerve parameters including CNFrD, and the assessment of nerve complexity may provide unique and distinct disease-specific topographical signatures. AI-driven pathways for the identification of neurodegeneration will facilitate earlier diagnosis and improved monitoring of the effectiveness of interventions.

## Figures and Tables

**Figure 1 jcm-11-06199-f001:**
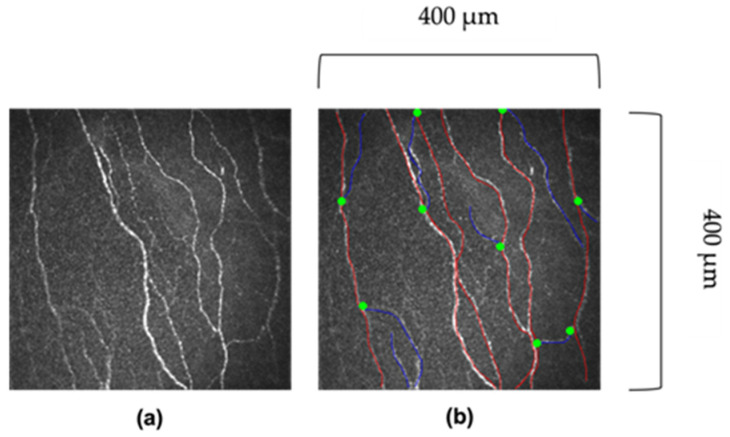
**Images of corneal nerves.** (**a**) Original image. (**b**) Red lines indicate the main corneal fibre, and blue lines indicate the corneal nerve branch. The greet dot indicates where the branch bifurcates off the main fibre. Images 400 μm × 400 μm.

**Figure 2 jcm-11-06199-f002:**
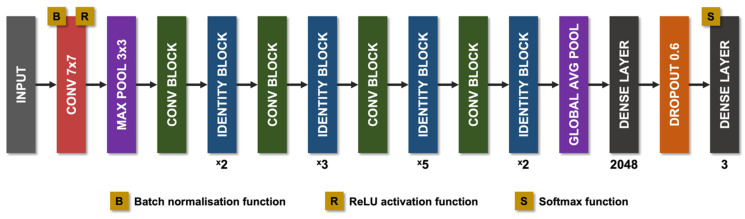
**Diagram of the modified ResNet-50 architecture used by Preston et al.** Pink rectangles correspond to convolutional layers, with the filter size given within. Purple rectangles corresponds to pooling layers, either maximum pool or global average pool. Green rectangles correspond to convolution blocks. Blue rectangles correspond to identity blocks. Dark grey rectangles correspond to dense layers. Orange rectangles correspond to dropout layers (dropout = 0.6). Avg, average; Conv, convolutional; Max, maximum; ReLU, rectified linear unit.

**Figure 3 jcm-11-06199-f003:**
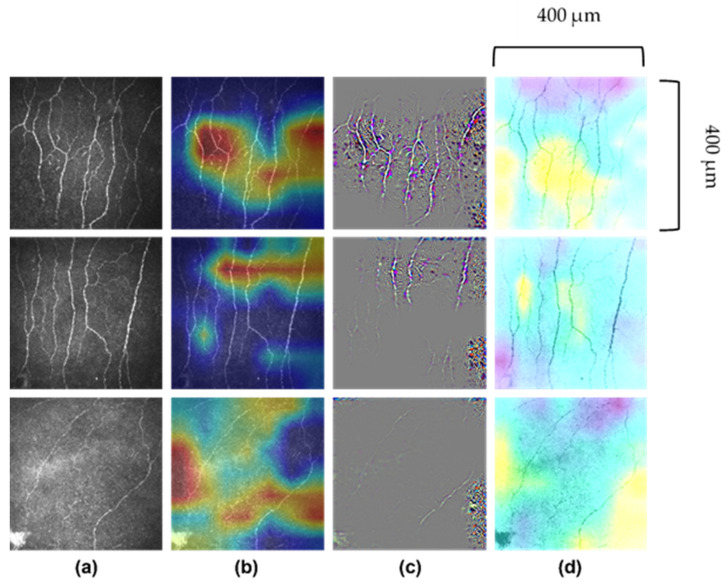
**Images of corneal nerves and their respective analysis by AI.** (**a**) Original images. (**b**) Attribution map generated by Grad-CAM (gradient-weighted-class activation mapping). (**c**) Attribution map generated by Guided Grad-CAM. (**d**) Attribution map generated by Occlusion Sensitivity. First row correctly predicted healthy volunteer participant; second row, correctly predicted participant with diabetes or prediabetes without neuropathy and third row, correctly predicted participant with diabetes or prediabetes with neuropathy. Images 400 μm × 400 μm.

**Table 1 jcm-11-06199-t001:** Manual and automated approaches using corneal confocal microscopy images to classify neuropathy.

Citation	Participants	Reference Standard	Index Test Threshold	Test and Target Condition	AUC	Sensitivity	Specificity
Diabetic Peripheral Neuropathy							
Perkins et al., 2018 [42]	Total = 998 T1D = 516 T2D = 482	Toronto Criteria Confirmed DPN	12.5 mm/mm^2^	Automated CNFL T1D—DPN	0.77	73	69
			12.3 mm/mm^2^	Automated CNFL T2D—DPN	0.68	69	63
			12.3 mm/mm^2^	Automated CNFL T1D and T2D—DPN	0.77	67	66
			Total < 8.6 mm/mm^2^	Automated CNFL T1D and T2D—DPN	-	88	88
Alam et al., 2017 [40]	T1D with neuropathy = 31 Control Participants = 27	Toronto Criteria Confirmed DPN	25 no/mm^2^ 36.5 no/mm^2^ 16.8 mm/mm^2^	CNFD—DPN	0.81	77	79
				CNBD—DPN	0.67	58	79
				CNFL—DPN	0.74	61	86
Chen et al., 2015 [41]	T1D = 63 Control = 26	Toronto Criteria Confirmed DPN	2 SD below the mean of the control group	Manual			
				CNFD—DPN	0.82	82	71
				CNFL—DPN	0.70	59	74
				CNBD—DPN	0.59	17	96
				Automated			
				CNFD—DPN	0.80	60	83
				CNFL—DPN	0.77	59	80
				CNBD—DPN	0.80	29	98
Edwards et al., 2014 [47]	DM = 231 Control = 61	Toronto Criteria Confirmed DPN	-	CNFL	0.64	32	87
			-	Tortuosity-standardised CNFL	0.67	38	88
Wang et al., 2021 [43]	Total = 220	Toronto Criteria Confirmed DPN	<15.3 mm/mm^2^	CNFL	0.70	80	59
	Control = 48		<39 no/mm^2^	CNBD	0.66	78	52
	T2D = 172		<25.68 n/mm^2^	CNFD	0.67	85	47
Other Peripheral Neuropathies							
Zhang et al., 2021 [48]	TTR-FAP = 15 Control = 15	Genetically Confirmed TTR-FAP	<17.99 mm/mm^2^	CNFL	0.88	80	93
			<21.95 mm/mm^2^	IWL	0.89	86	80
Central Peripheral Neuropathies							
Che et al., 2021 [49]	Total = 82	Clinically confirmed PD	<10.08 mm/mm^2^	CNFL	0.67	85	45
	PD = 42		<22.85 n/mm^2^	CNFD	0.96	95	88
	Control = 40		<26.72 n/mm^2^	CNBD	0.69	92	52
Fernandes et al., 2021 [50]	Total = 82 MS = 60 Control = 22	Clinically confirmed MS	-	CNFD	0.84	-	-
			-	CNBD	0.84	-	-
			-	CNFL	0.74	-	-
			-	CNFT	0.72	-	-

AUC—area under the curve; CNBD—corneal nerve fibre branch density; CNFD—corneal nerve fibre density; CNFL—corneal nerve fibre length; CNFT—corneal nerve fibre tortuosity; DM—diabetes mellitus; DPN—diabetic peripheral neuropathy; IWL—inferior whorl length; PD—Parkinson’s disease; PMNAP—peroneal motor nerve amplitude; PMNCV—peroneal motor nerve conduction velocity; n—number; MS—multiple sclerosis; SNAP—sensory nerve action potential; SNCV—sural nerve conduction velocity; T1D—type 1 diabetes; T2D—type 2 diabetes; TTR-FAP—transthyretin familial amyloid polyneuropathy.

**Table 2 jcm-11-06199-t002:** Corneal confocal microscopy (CCM) biomarkers.

Parameter	Description	Unit of Measurement
Corneal nerve fibre length (CNFL)	Length of all main nerve fibres and branches	mm/mm^2^
Corneal nerve fibre density (CNFD)	Number of main nerve fibres	no/mm^2^
Corneal nerve branch density (CNBD)	Number of main nerve fibre branches	no/mm^2^

**Table 3 jcm-11-06199-t003:** Artificial intelligence approaches using corneal confocal microscopy images to classify neuropathy.

Citation	Participants	No. of Images	Study Methodology	Population	AUC	Sensitivity	Specificity	Classification Accuracy	Results Summary
Scarpa et al., 2019 and Scarpa et al., 2020 [89,91]	Total = 100 DPN = 50 Control = 50	Total = 600; Training = 480; Cross-validation = 600; Evaluation = 120	CNN	Neuropathy vs. Control (single block)	-	98	96	97	CNN identifies ROI allowing multiple images to be binarised into two separate categories demonstrating diagnostic efficacy
				Neuropathy vs. Control (whole subject)	-	98	94	96	
Williams et al., 2020 [90]	Total = 222 DPN+ve = 132 DPN-ve = 90	Images used for training the Liverpool CNN Total = 1698;	CNN and DLA	DPN+ve vs. DPN-ve	0.83	68	87	-	The Liverpool CNN and Liverpool DLA can quantify corneal nerve morphometrics in participants with confirmed DPN demonstrating diagnostic efficacy
		External validation of the CNN/DLA Total =1578; Images evaluated using the Liverpool CNN/DLA;							
		Participants with and without DPN as per the Toronto expert criteria included. Total images = 2137							
Salahouddin et al., 2021 [130]	Total = 108 Control = 21 DPN+ve = 25 DPN−ve = 62	Training = 174; Validation = 534	DL ANFIS	DPN−ve cs Control	0.86 (0.77–0.94)	84	71	-	Based on CCM images alone ANFIS classified 43% of participants as DPN+ve demonstrating diagnostic utility
				DPN−ve vs. DPN+ve	0.95 (0.91–0.99)	92	80	-	
				Control vs. DPN+ve	1.0 (0.99–1.0)	100	95	-	
Preston et al., 2022 [92]	Total = 369 Control = 90 DPN+ve = 130 DPN−ve = 149	Training = 245; Validation = 84; Test = 40	DLA	Control	-	100	-	100	Based on a single CCM image without pre-processing DLA can faithfully classify participants into controls, DPN+ve and DPN−ve categories demonstrating diagnostic utility and accuracy
				DPN-ve	-	85	-	85	
				DPN+ve	-	83	-	83	

AUC—area under the curve; ANFIS—adaptive neurofuzzy inference system; CCM—corneal confocal microscopy; CNN—convolutional neural network; DL—deep learning; DLA; deep learning algorithm; DPN−ve—no diabetic peripheral neuropathy; DPN+ve—diabetic peripheral neuropathy; ROI—region of interest.

## Data Availability

Not applicable.

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
