# Peer review of "Artificial Intelligence and Corneal Confocal Microscopy: The Start of a Beautiful Relationship"

_jcm, 2022, doi:10.3390/jcm11206199_

Round 1

Reviewer 1 Report

The paper has an interesting topic based on its title which is: Artificial Intelligence and Corneal Confocal Microscopy: the start of a beautiful relationship

However, the paper makes a great review of the current state of the use of confocal microscopy for several neuropathies, and a poor explanation of the main topic as is Artifial intelligence (AI) and the aplication in Confocal Microscopy images. I would recommend giving more importance to the topic of AI and resume the whole introduction much more.

The use of so many acronyms makes dificult the understanding. There are some that can be avoided, as they are only used 2 times.

For example, the acronym IENFD can be perfectly replaced by intraepithelial biopsy.

I will recomend to the authors focused in the main topic with a little review of the use of confocal microscopy.

Reviewer 2 Report

Authors made big efforts in analyzing the state of the art of AI and CCM, studying and properly citing the current literature in the field. 

The manuscript is well written and comprensible. However, I think that small changes could make the review more appealing and comprensible for readers. Indeed, tables are not very clear at first view and the paper lacks of schemes and figures.

In particular: 

- make a list of acronimous

- add scalebars in figure 2 

- format tables in order to organize clear rows and columns

- add figures and schemes, for example where the corneal anatomy is explained and where the parameters consider for the image analysis are listed

Reviewer 3 Report

Do you have any suggestion how to obtain a big data including CCM or how to standardiseCCM automated analysis?

Round 2

Reviewer 1 Report

The paper is improved